# Learning Probabilistic Topological Represen­tations Using Discrete Morse Theory

**Xiaoling Hu** [*]
Stony Brook University

**Dimitris Samaras**
Stony Brook University

**Chao Chen**
Stony Brook University

## Abstract

Accurate delineation of fine-scale structures is a very important yet challenging problem. Existing methods use topological information as an additional training loss, but are ultimately making pixel-wise predictions. In this paper, we propose a novel deep learning based method to learn topological/structural [1] representations. We use discrete Morse theory and persistent homology to construct a one-parameter family of structures as the topological/structural representation space. Furthermore, we learn a probabilistic model that can perform inference tasks in such a topolog­ical/structural representation space. Our method generates true structures rather than pixel-maps, leading to better topological integrity in automatic segmentation tasks. It also facilitates semi-automatic interactive annotation/proofreading via the sampling of structures and structure-aware uncertainty.

## 1 Introduction

Accurate segmentation of fine-scale structures, e.g., vessels, neurons and membranes is crucial for downstream analysis. In recent years, topology-inspired losses have been proposed to improve structural accuracy (Hu et al., 2019; 2021; Shit et al., 2021; Mosinska et al., 2018; Clough et al., 2020). These losses identify topologically critical locations at which a segmentation network is error-prone, and force the network to improve its prediction at these critical locations.

However, these loss-based methods are still not ideal. They are based on a standard segmentation network, and thus *only learn pixel-wise feature representations*. This causes several issues. First, a standard segmentation network makes pixel-wise predictions. Thus, at the inference stage, topological errors, e.g. broken connections, can still happen, even though they may be mitigated by the topology-inspired losses. Another issue is in uncertainty estimation, i.e., estimating how certain a segmentation network is at different locations. Uncertainty maps can direct the focus of human annotators for efficient proofreading. However, for fine-scale structures, existing pixel-wise uncertainty maps are not effective. As shown in Fig. 1(d), every pixel adjacent to a vessel branch is highly uncertain, in spite of whether the branch is salient or not. What is more desirable is a structural uncertainty map that can highlight uncertain branches (e.g., Fig. 1(f)).

To fundamentally address these issues, we propose to directly model and reason about the structures. In this paper, we propose *a novel deep learning based method that directly learns the topologi­cal/structural representation of images*. To move from pixel space to structure space, we apply the classic discrete Morse theory (Milnor, 1963; Forman, 2002) to decompose an image into a Morse complex, consisting of structural elements like branches, patches, etc. These structural elements are the hypothetical structures one can infer from the input image. Their combinations constitute a space of structures arising from the input image. See Fig. 2(c) for an illustration.

For further reasoning with structures, we propose to learn a probabilistic model over the structure space. The challenge is that the space consists of exponentially many branches and is thus of very high dimension. To reduce the learning burden, we introduce the theory of persistent homology (Sousbie, 2011; Delgado-Friedrichs et al., 2015; Wang et al., 2015) for structure pruning. Each branch has its own persistence measuring its relative saliency. By continuously thresholding the complete Morse complex in terms of persistence, we obtain a sequence of Morse complexes parameterized by the persistence threshold, $\epsilon$. See Fig. 2(d). By learning a Gaussian over $\epsilon$, we learn a parametric probabilistic model over these structures.

---

[*]Email: Xiaoling Hu (xiaolhu@cs.stonybrook.edu).

[1]We will be using the terms topology/topological and structure/structural interchangeably in this paper.

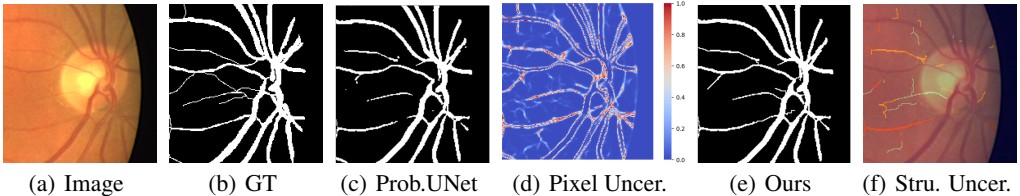

| (a) Image | (b) GT | (c) Prob.UNet | (d) Pixel Uncer. | (e) Ours | (f) Stru. Uncer. |

Figure 1: Illustration of structural segmentation and structure-level uncertainty. Compared with Probabilistic-UNet (Kohl et al., 2018) (Fig. 1(c)-(d)), the proposed method is able to generate structure-preserving segmentation map (Fig. 1(e)), and structure-level uncertainty (Fig. 1(f)).

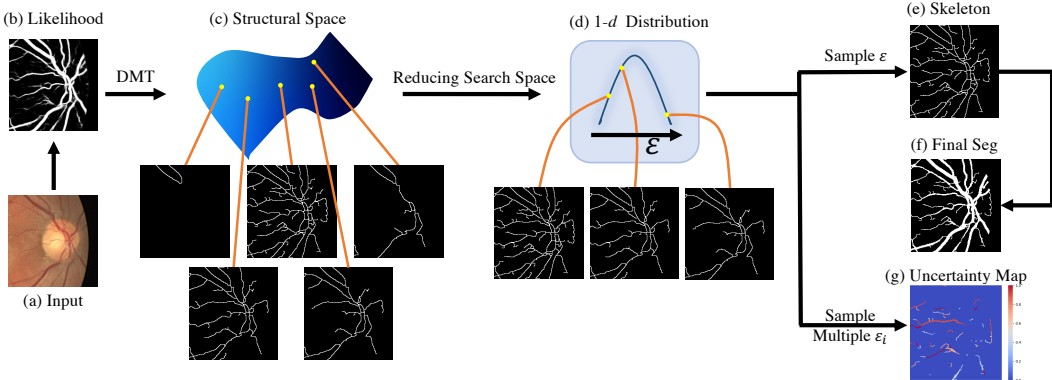

Figure 2: The probabilistic topological/structural representation. **(a)** is a sample input, **(b)** is the predicted likelihood map from the deep neural network, **(c)** is the whole structure space obtained by running a discrete Morse theory algorithm on the likelihood map, **(d)** the 1-$d$ structural family parametrized by the persistence threshold $\epsilon$, as well as a Gaussian distribution over $\epsilon$, **(e)** a sampled skeleton, **(f)** the final structural segmentation map generated using the skeleton sample, and **(g)** the uncertainty map generated by multiple segmentations.

This parametric probabilistic model over structure space allows us to make direct structural predictions via sampling (Fig. 2(e)), and to estimate the empirical structure-level uncertainty via sampling (Fig. 2(g)). The benefit is two-fold: First, direct prediction of structures will ensure the model outputs always have structural integrity, even at the inference stage. This is illustrated in Fig. 1(e). Samples from the probabilistic model are all feasible structural hypotheses based on the input image, with certain variations at uncertain locations. This is in contrast to state-of-the-art methods using pixel-wise representations (Fig. 1(c)-(d)). Note the original output structure (Fig. 2(e), also called skeleton) is only 1-pixel wide and may not serve as a good segmentation output. In the inference stage, we use a post-processing step to grow the structures without changing topology as the final segmentation prediction (Fig. 2(f)). More details are provided in Sec. 3.2 and Fig. 5.

Second, the probabilistic structural model can be seamlessly incorporated into semi-automatic interactive annotation/proofreading workflows to facilitate large scale annotation of these complex structures (see Fig. 5). This is especially important in the biomedical domain where fine-scale structures are notoriously difficult to annotate, due to the complex 2D/3D morphology and low contrast near extremely thin structures. Our probabilistic model makes it possible to identify uncertain structures for efficient interactive annotation/proofreading. Note that the structure space is crucial for uncertainty reasoning. As shown in Fig. 1(f) and Fig. 2(g), our structural uncertainty map highlights uncertain branches for efficient proofreading. On the contrary, traditional pixel-wise uncertainty map (Fig. 1(d)) is not helpful at all; it highlights all pixels on the boundary of a branch.

The main contributions of this paper are:

1. We propose a novel deep learning method which learns structural representations, based on discrete Morse theory and persistent homology.
2. We learn a probabilistic model over the structure space, which facilitates different tasks such as topology-aware segmentation, uncertainty estimation and interactive proofreading.
3. We validate our method on various datasets with *rich and complex structures*. It outperforms state-of-the-art methods in both deterministic and probabilistic categories.

## 2    RELATED WORK

**Structure/Topology-aware deep image segmentation.** A number of recent works have tried to segment with correct topology with additional topology-aware losses (Mosinska et al., 2018; Hu et al., 2019; Clough et al., 2020; Hu et al., 2021; Shit et al., 2021). Specifically, UNet-VGG (Mosinska et al., 2018) detects linear structures with pretrained filters. clDice (Shit et al., 2021) introduces an additional Dice loss for extracted skeleton structures. TopoLoss (Hu et al., 2019; Clough et al., 2020) learns to segment with correct topology explicitly with a differentiable loss by leveraging the concept of persistent homology. Similarly, DMT-loss (Hu et al., 2021) tries to identify the topological critical structures via discrete Morse theory, and then force the network to make correct pixel-wise prediction on these structures. All these losses, although aiming at topological integrity, still cannot change the pixel-wise prediction nature of the backbone network. To the best of our knowledge, no existing methods generate structural representations/predictions like our proposed method.

Additionally, the discrete Morse complex has been used for image analysis, but only as a preprocessing step (Delgado-Friedrichs et al., 2015; Robins et al., 2011; Wang et al., 2015; Dey et al., 2019) or as a conditional input of a neural network (Banerjee et al., 2020).

**Segmentation uncertainty.** Uncertainty estimation has been the focus of research in recent years (Graves, 2011; Gal & Ghahramani, 2016; Lakshminarayanan et al., 2017; Moon et al., 2020). However, most existing work focuses on the classification problem. In terms of image segmentation, the research is still relatively limited. Some existing methods directly apply classification uncertainty to individual pixels, e.g., dropout (Kendall et al., 2015; Kendall & Gal, 2017). This, however, is not taking into consideration the image structures. Several methods estimate the uncertainty by generating an ensemble of segmentation (Lakshminarayanan et al., 2017) or using multi-heads (Rupprecht et al., 2017; Ilg et al., 2018). Notably, Probabilistic-UNet (Kohl et al., 2018) learns a distribution over the latent space and then samples over the latent space to produce segmentation samples. When it comes to uncertainty, however, these methods can still only generate a pixel-wise uncertainty map, using the frequency of appearance of each pixel in the sample segmentations. These methods are fundamentally different from ours, which makes predictions on the structures.

## 3    METHOD

Our key innovation is to restructure the output of a neural network so that it is indeed making predictions over a space of structures. This is achieved through insights into the topological structures of an image and the usage of several important tools in topological data analysis.

To move from pixel space to structure space, we apply discrete Morse theory to decompose an image into a Morse complex, consisting of structures like branches, patches, etc. For simplification, we will use the term "branch" to denote a single piece of Morse structure. These Morse branches are the hypothetical structures one can infer from the input image. This decomposition is based on a likelihood function produced by a pixel-wise segmentation network trained in parallel. Thus it is of a good quality, i.e., the structures are close enough to the true structures.

Any binary labeling of these Morse branches is a legitimate segmentation; we call it a *structural segmentation*. But for full-scope reasoning of the structure space, instead of classifying these branches one-by-one, we would like to have the full inference, i.e., predicting a probability distribution for each branch. To further reduce the degrees of freedom to make the inference easier, we apply persistent homology to filter these branches with regard to their saliency. This gives us a linear size family of structural segmentations, parameterized by a threshold $\epsilon$. Finally, we learn a 1D Gaussian distribution for the $\epsilon$ as our probabilistic model. This gives us the opportunity not only to sample segmentations, but also to provide a probability for each branch, which can be useful in downstream tasks including proofreading. In Sec. 3.1, we introduce the discrete Morse theory and how to construct the space of Morse structures. We also explain how to use persistent homology to reduce the search space of reasoning into a 1-parameter family. In Sec. 3.2, we will provide details on how our deep neural network is constructed to learn the probabilistic model over the structure space, as illustrated in Fig. 4.

### 3.1    CONSTRUCTING THE STRUCTURE SPACE

In this section, we explain how to construct a structural representation space using discrete Morse theory. The resulting structural representation space will be used to build a probabilistic model. We will then discuss how to reduce the structure space into a 1-parameter family of structural

segmentations, using persistent homology. We assume a 2D input image, although the method naturally extends to 3D images.

Given a reasonably clean input (e.g., the likelihood map of a deep neural network, Fig. 2(b)), we treat the 2D likelihood as a terrain function, and Morse theory (Milnor, 1963) can help to capture the structures regardless of weak/blurred conditions. See Fig. 3 for an illustration. The weak part of a line in the continuous map can be viewed as the local dip in the mountain ridge of the terrain. In the language of Morse theory, the lowest point of this dip is a saddle point ($S$ in Fig. 3(b)), and the mountain ridges which are connected to the saddle point ($M_1S$ and $M_2S$) are called the stable manifolds of the saddle point.

We mainly focus on 2D images in this paper. We consider two dimensional continuous function $f : \mathbb{R}^2 \to \mathbb{R}$. For a point $x \in \mathbb{R}^2$, the gradient can be computed as $\nabla f(x) = [\frac{\partial f}{\partial x_1}, \frac{\partial f}{\partial x_2}]^T$. We call a point $x = (x_1, x_2)$ *critical* if $\nabla f(x) = 0$. For a Morse function defined on $\mathbb{R}^2$, a critical point could be a minimum, a saddle or a maximum.

Consider a continuous line (the red rectangle region in Fig. 3(a)) in a 2D likelihood map. Imagine if we put a ball on one point of the line, then $-\nabla f(x)$ indicates the direction which the ball will flow down. By definition, the ball will eventually flow to the critical points where $\nabla f(x) = 0$. The collection of points whose ball eventually flows to $p$ ($\nabla f(p) = 0$) is defined as the stable manifold (denoted as $S(p)$) of point $p$. Intuitively, for a 2D function $f$, the stable manifold $S(p)$ of a minimum $p$ is the entire valley of $p$ (similar to the watershed algorithm); similarly, the stable manifold $S(q)$ of a saddle point $q$ consists of the whole ridge line which connects two local maxima and goes through the saddle point. See Fig. 3(b) as an illustration.

For a link-like structure, the stable manifold of a saddle contains the topological structures (usually curvilinear) of the continuous likelihood map predicted by deep neural networks, and they are exactly what we want to recover from noisy images. In practice, we adopt the discrete version of Morse theory for images.

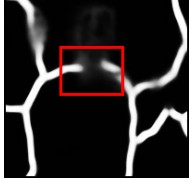 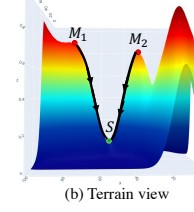 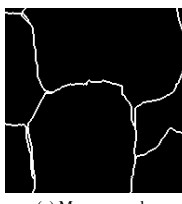

(a) Likelihood map     (b) Terrain view     (c) Morse complex

Figure 3: **(a)** shows a sample likelihood map from the deep neural network, and **(b)** is the terrain view of the red patch in **(a)** and illustrates the stable manifold of a saddle point in 2D case for a line-like structure. **(c)** is the 2D Morse complex generated by DMT from **(a)**.

**Discrete Morse theory.** Take a 2D image as a 2-dimensional cubical complex (Wagner et al., 2012; Kaczynski et al., 2004). A 2-dimensional cubical complex then contains 0-, 1-, and 2-dimensional cells, which correspond to vertices (pixels), edges and squares, respectively. In the setting of discrete Morse theory (DMT) (Forman, 1998; 2002), a pair of adjacent cells, termed as discrete gradient vectors, compose the gradient vector. Critical points ($\nabla f(x) = 0$) are those critical cells which are not in any discrete gradient vectors. In the 2D domain, a minimum, a saddle and a maximum correspond to a critical vertex, a critical edge and a critical square respectively. A 1-stable manifold (the stable manifold of a saddle point) in 2D corresponds to a *V-path*, i.e., connecting two local maxima and a saddle. See Fig. 3(b). And the Morse complex generated by the DMT algorithm is illustrated in Fig. 3(c).

**Constructing the full structure space.** In this way, by using discrete Morse theory, for a likelihood map from the deep neural network, we can extract all the stable manifolds of the saddles, whose compositions constitute the full structure space. *Formally, we call any combinations of these stable manifolds a structure.* Fig. 2(c) illustrates 5 different structures. This structure space, however, is of exponential size. Assume there are $N$ pieces of stable manifolds/branches for a given likelihood map. Any combinations of these stable manifolds/branches will be a potential structure. We will have $2^N$ possible structures in total. This can be computationally prohibitive to construct and to model. We need a principled way to reduce the structural search space.

**Reducing the structural search space with persistent homology.** We propose to use persistent homology (Sousbie, 2011; Delgado-Friedrichs et al., 2015; Wang et al., 2015) to reduce the structural search space. Persistent homology is an important tool for topological data analysis (Edelsbrunner & Harer, 2022; Edelsbrunner et al., 2000). Intuitively, we grow a Morse complex by gradually including more and more discrete elements (called cells) from empty. A branch of the Morse complex is a special

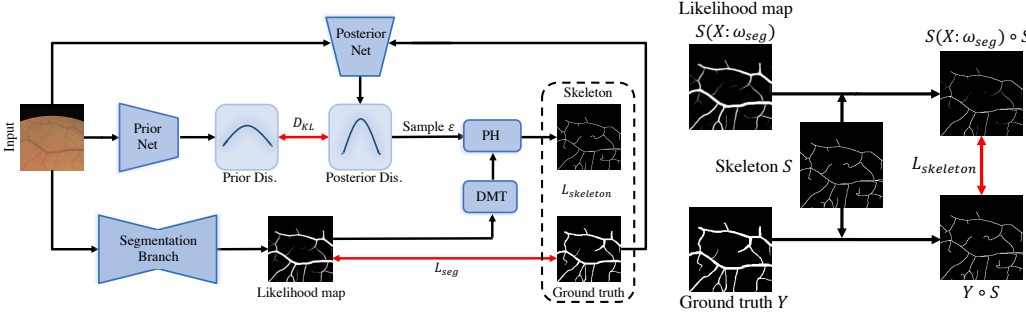

(a) Overview of the proposed framework      (b) Details of $L_{skeleton}$

Figure 4: The overall workflow of the training stage. The red arrows indicate supervision.

type of cell. Other types include vertices, patches, etc. Cells will be continuously added to the complex. New branches will be born and existing branches will die. The persistence algorithm (Edelsbrunner et al., 2000) pairs up all these critical cells as birth and death pairs. The difference of their function values is essentially the life time of the specific topological structure/branch, which is called the *persistence*. The importance of a branch is associated with its persistence. Intuitively, the longer the persistence of a specific branch is, the more important the branch is.

Recall that our original construction of the structure space considers all possible combinations of branches, and thus can have exponentially many combinations. Instead, we propose to only select branches with high persistence as important ones. By doing this, we will be able to prune the less important/noisy branches very efficiently, and recover the branches with true signals. Specifically, the structure pruning is done via *Morse cancellation* (more details are included in Appendix A.3) operation. The persistence thresholding provides us the opportunity to obtain a *structure space of linear size*. We start with the complete Morse complex, and continuously increase the threshold $\epsilon$. At each threshold, we obtain a structure by filtering with $\epsilon$ and only keeping the branches whose persistence is above $\epsilon$. This gives a sequence of structures parametrized by $\epsilon$. As shown in Fig. 2(d), the family of structures represents different structural densities.

The one-parameter space allows us to easily learn a probabilistic model and carry out various inference tasks such as segmentation, sampling, uncertainty estimation and interactive proofreading. Specifically, we will learn a Gaussian distribution over the persistence threshold $\epsilon$, $\epsilon \sim N(\mu, \sigma)$. Denote the persistence of a branch $b$ as $\epsilon_b$. Any branch $b$ belongs to the structure map $M$ (we also call the structure map $M$ a structural segmentation) as long as its persistence is higher or equal to the persistence threshold of $M$, i.e., $b \in M$ if and only if $\epsilon_b \geq \epsilon_M$, where $\epsilon_M$ is used to generate $M$. More details will be provided in Sec. 3.2.

**Approximation of Morse structures for volume data.** In the 2D setting, the stable manifold of saddles is composed of curvilinear structures, and the captured Morse structures will essentially contain the *non-boundary edges*, which fits well with vessel data. However, the output structures should always be *boundary edges* for volume data, which cannot be dealt with the original discrete Morse theory. Consequently, we approximate the Morse structures of 2D volume data with the boundaries of the stable manifolds of local minima. As mentioned above, the stable manifold of a local minimum $p$ in the 2D setting corresponds to the whole valley, and the boundaries of these valleys construct the approximation of the Morse structures for volume data. Similar to the original discrete Morse theory, we also introduce a persistence threshold parameter $\epsilon$ and use persistent homology to prune the less important branches. The details of the proposed persistent-homology filtered topology watershed algorithm are illustrated in Appendix A.4.

## 3.2 NEURAL NETWORK ARCHITECTURE

In this section, we introduce our neural network that learns a probabilistic model over the structural representation to obtain structural segmentations. See Fig. 4 for an illustration of the overall pipeline.

Since the structural reasoning needs a sufficiently clean input to construct discrete Morse complexes, our method first obtains such a likelihood map by training a segmentation branch which is supervised by the standard segmentation loss, binary cross-entropy loss. Formally, $L_{seg} = L_{bce}(Y, S(X; \omega_{seg}))$, in which $X$ is the input image, $\omega_{seg}$ is the segmentation branch's weight, $S(X; \omega_{seg})$ is the output likelihood map, and $Y$ is the ground truth.

The output likelihood map, $S(X; \omega_{seg})$, is used as the input for the discrete Morse theory algorithm (DMT), which generates a discrete Morse complex consisting of all possible Morse branches from the likelihood map. Thresholding these branches using persistent homology with different $\epsilon$ values will produce different structures. We refer to the DMT computation and the persistent homology thresholding operation as $f_{DMT}$ and $f_{PH}$. So given a likelihood map $S(X; \omega_{seg})$ and a threshold $\epsilon$, we can generate a structure (which we call a skeleton): $S_{skeleton}(\epsilon) = f_{PH}(f_{DMT}(S(X; \omega_{seg})); \epsilon)$. Next, we discuss how to learn the probabilistic model. Recall that we want to learn a Gaussian distribution over the persistent homology threshold, $\epsilon \sim N(\mu, \sigma)$. The parameters $\mu$ and $\sigma$ are learned by a neural network called the *posterior network*. The network uses the input image $X$ and the corresponding ground truth mask $Y$ as input, and outputs the parameters $\mu(X, Y; \omega_{post})$ and $\sigma(X, Y; \omega_{post})$. $\omega_{post}$ is the parameter of the network.

During training, at each iteration, we draw a sample $\epsilon$ from the distribution ($\epsilon \sim N(\mu, \sigma)$). Using the sample $\epsilon$, together with the likelihood map, we can generate the corresponding sampled structure, $S_{skeleton}(\epsilon)$. This skeleton will be compared with the ground truth for supervision. To compare a sampled skeleton, $S_{skeleton}(\epsilon)$, with ground truth $Y$, we use the skeleton to mask both $Y$ and the likelihood map $S(X; \omega_{seg})$, and then compare the skeleton-masked ground truth and the likelihood using binary cross-entropy loss: $L_{bce}(Y \circ S_{skeleton}(\epsilon), S(X; \omega_{seg}) \circ S_{skeleton}(\epsilon))$, in which $\circ$ denotes the Hadamard product.

To learn the distribution, we use the expected loss:

$$L_{skeleton} = \mathbb{E}_{\epsilon \sim N(\mu, \sigma)} L_{bce}(Y \circ S_{skeleton}(\epsilon), S(X; \omega_{seg}) \circ S_{skeleton}(\epsilon)) \tag{1}$$

We backpropagate this loss through the posterior network using the reparameterization technique in (Kingma & Welling, 2013). More details are provided in Appendix A.5. Note that this loss will also provide supervision to the segmentation network through the likelihood map.

**Learning a prior network from the posterior network.** Although our posterior network can learn the distribution well, it relies on the ground truth mask $Y$ as input. This is not available at inference stage. To address this issue, inspired by VAE (Kingma & Welling, 2013; Kohl et al., 2018), we use another network to learn the distribution of $\epsilon$ with only the image $X$ as input. We call this network the *prior net*. We denote by $P$ the distribution using parameters predicted by the prior network, and denote by $Q$ the distribution predicted by the posterior network.

During training, we want to force the prior net to mimic the posterior net; and then in the inference stage, we use the prior net to obtain a reliable distribution over $\epsilon$ with only the image $X$. Thus, we incorporate the Kullback-Leibler divergence of these two distributions, $D_{KL}(Q||P) = \mathbb{E}_{\epsilon \sim Q}(\log \frac{Q}{P})$, which measures how close the prior distribution $P$ ($N(\mu_{prior}, \sigma_{prior})$) is to the posterior distribution $Q$ ($N(\mu_{post}, \sigma_{post})$).

**Training the neural network.** The final loss is composed by the standard segmentation loss, the skeleton loss $L_{skeleton}$, and the KL divergence loss, with two hyperparameters $\alpha$ and $\beta$ to balance the three terms,

$$L(X, Y) = L_{seg} + \alpha L_{skeleton} + \beta D_{KL}(Q||P) \tag{2}$$

The network is trained to jointly optimize the segmentation branch and the probabilistic branch (containing both prior and posterior nets) simultaneously. During the training stage, the KL divergence loss ($D_{KL}$) pushes the prior distribution towards the posterior distribution. The training scheme is also illustrated in Fig. 4.

**Inference stage: generating structure-preserving segmentation maps.** In the inference stage, given an input image, we are able to produce unlimited number of plausible structure-preserving skeletons via sampling. We use a postprocessing step to grow the 1-pixel wide structures/skeletons without changing their topology as the final structural segmentation. Specifically, the skeletons are overlaid on the binarized initial segmentation map (Fig. 5(c)), and only the connected components which exist in the skeletons are kept as the final segmentation maps (Fig. 5(e)). In this way, each plausible skeleton (Fig. 5(d)) generates one final segmentation map (Fig. 5(e)) and it has exact the same topology as the corresponding skeleton. The pipeline of the procedure is illustrated in Fig. 5.

**Uncertainty of structures.** Given a learned prior distribution, $P$, over the family of structural segmentations, we can naturally calculate the probability of each Morse structure/branch. Recall a branch $b$ has its persistence $\epsilon_b$. And the prior probability of a structural segmentation map $M$ is $P(\epsilon_M)$, in which $\epsilon_M$ is used to generate $M$. Also any branch $b$ whose persistence is higher

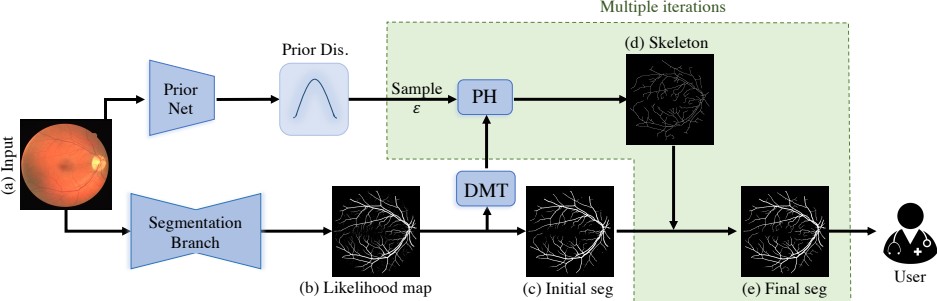

Figure 5: The inference and interactive annotation/proofreading pipeline.

or equal to the persistence threshold of $M$ belongs to $M$, i.e., $b \in M$ if and only if $\epsilon_b \geq \epsilon_M$. Therefore, the probability of a branch $b$ being in a segmentation map $M$ such as $\epsilon_M \sim P$ follows a Bernoulli distribution with the probability $Pr(b)$ being the cumulative distribution function (CDF) of $P$, $CDF_P(\epsilon_b) = P(\epsilon \leq \epsilon_b)$. This can be directly calculated at inference, and the absolute difference of the CDF from 0.5 is the confidence (which equals 1-uncertainty) of the Morse structure/branch $b$. We add an illustration of uncertainty estimation for branches in Fig. 14 (Sec. A.12 of Appendix).

## 4 EXPERIMENTS

Our method directly makes prediction and inference on structures rather than on pixels. This can significantly benefit downstream tasks. While probabilities of structures can certainly be used for further analysis of the structural system, in this paper we focus on both automatic image segmentation and semi-automatic annotation/proofreading tasks. On automatic image segmentation, we show that direct prediction can ensure topological integrity even better than previous topology-aware losses. This is not surprising as our prediction is on structures. On semi-automatic proofreading task, we show our structure-level uncertainty can assist human annotators to obtain satisfying segmentation annotations in a much more efficient manner than previous methods.

### 4.1 AUTOMATIC TOPOLOGY-AWARE IMAGE SEGMENTATION

**Datasets.** We use three datasets to validate the efficacy of the proposed method: **ISBI13** (Arganda-Carreras et al., 2013) (volume), **CREMI** (volume), and **DRIVE** (Staal et al., 2004) (vessel). More details are included in Appendix A.6.

**Evaluation metrics.** We use four different evaluation metrics: **Dice score**, **ARI**, **VOI**, and **Betti number error**. Dice is a popular pixel-wise segmentation metric, and the other three are structure/topology-aware segmentation metrics. More details are included in Appendix A.7.

**Baselines.** We compare the proposed method with two kinds of baselines: 1) Standard segmentation baselines: **DIVE** (Fakhry et al., 2016), **UNet** (Ronneberger et al., 2015), **UNet-VGG** (Mosinska et al., 2018), **TopoLoss** (Hu et al., 2019) and **DMT** (Hu et al., 2021). 2) Probabilistic-based segmentation methods: **Dropout UNet** (Kendall et al., 2015) and **Probabilistic-UNet** (Kohl et al., 2018). More details about these baselines are included in Appendix A.8.

**Quantitative and qualitative results.** Table 1 shows the quantitative results comparing to several baselines. Note that for deterministic methods, the numbers are computed directly based on the outputs; while for probabilistic methods, we generate five segmentation masks and report the averaged numbers over the five segmentation masks for each image (for both the baselines and the proposed method). We use t-test (95% confidence interval) to determine the statistical significance and highlight the significant better results. From the table, we can observe that the proposed method achieves significant better performances in terms of topology-aware metrics (ARI, VOI and Betti Error).

Fig. 6 shows qualitative results. Comparing with DMT (Hu et al., 2021), our method is able to produce a set of true structure-preserving segmentation maps, as illustrated in Fig. 6(e-g). Note that compared with the existing topology-aware segmentation methods, our method is more capable of recovering the weak connections by using Morse skeletons as hints. More qualitative results are included in Appendix A.2.

Table 1: Quantitative results for different models on three different biomedical datasets.

| Method | Dice ↑ | ARI ↑ | VOI ↓ | Betti Error ↓ |
|---|---|---|---|---|
| | ISBI13 (Volume) | | | |
| DIVE | $0.9658 \pm 0.0020$ | $0.6923 \pm 0.0134$ | $2.790 \pm 0.025$ | $3.875 \pm 0.326$ |
| UNet | $0.9649 \pm 0.0057$ | $0.7031 \pm 0.0256$ | $2.583 \pm 0.078$ | $3.463 \pm 0.435$ |
| UNet-VGG | $0.9623 \pm 0.0047$ | $0.7483 \pm 0.0367$ | $1.534 \pm 0.063$ | $2.952 \pm 0.379$ |
| TopoLoss | $0.9689 \pm 0.0026$ | $0.8064 \pm 0.0112$ | $1.436 \pm 0.008$ | $1.253 \pm 0.172$ |
| DMT | $\mathbf{0.9712 \pm 0.0047}$ | $0.8289 \pm 0.0189$ | $1.176 \pm 0.052$ | $1.102 \pm 0.203$ |
| Dropout UNet | $0.9591 \pm 0.0031$ | $0.7127 \pm 0.0181$ | $2.483 \pm 0.046$ | $3.189 \pm 0.371$ |
| Prob.-UNet | $0.9618 \pm 0.0019$ | $0.7091 \pm 0.0201$ | $2.319 \pm 0.041$ | $3.019 \pm 0.233$ |
| **Ours** | $0.9637 \pm 0.0032$ | $\mathbf{0.8417 \pm 0.0114}$ | $\mathbf{1.013 \pm 0.081}$ | $\mathbf{0.972 \pm 0.141}$ |
| | CREMI (Volume) | | | |
| DIVE | $0.9542 \pm 0.0037$ | $0.6532 \pm 0.0247$ | $2.513 \pm 0.047$ | $4.378 \pm 0.152$ |
| UNet | $0.9523 \pm 0.0049$ | $0.6723 \pm 0.0312$ | $2.346 \pm 0.105$ | $3.016 \pm 0.253$ |
| UNet-VGG | $0.9489 \pm 0.0053$ | $0.7853 \pm 0.0281$ | $1.623 \pm 0.083$ | $1.973 \pm 0.310$ |
| TopoLoss | $0.9596 \pm 0.0029$ | $0.8083 \pm 0.0104$ | $1.462 \pm 0.028$ | $1.113 \pm 0.224$ |
| DMT | $0.9653 \pm 0.0019$ | $0.8203 \pm 0.0147$ | $1.089 \pm 0.061$ | $0.982 \pm 0.179$ |
| Dropout UNet | $0.9518 \pm 0.0018$ | $0.6814 \pm 0.0202$ | $2.195 \pm 0.087$ | $3.190 \pm 0.198$ |
| Prob.-UNet | $0.9531 \pm 0.0022$ | $0.6961 \pm 0.0115$ | $1.901 \pm 0.107$ | $2.931 \pm 0.177$ |
| **Ours** | $\mathbf{0.9681 \pm 0.0016}$ | $\mathbf{0.8475 \pm 0.0043}$ | $\mathbf{0.935 \pm 0.069}$ | $\mathbf{0.919 \pm 0.059}$ |
| | DRIVE (Vessel) | | | |
| DIVE | $0.7543 \pm 0.0008$ | $0.8407 \pm 0.0257$ | $1.936 \pm 0.127$ | $3.276 \pm 0.642$ |
| UNet | $0.7491 \pm 0.0027$ | $0.8343 \pm 0.0413$ | $1.975 \pm 0.046$ | $3.643 \pm 0.536$ |
| UNet-VGG | $0.7218 \pm 0.0013$ | $0.8870 \pm 0.0386$ | $1.167 \pm 0.026$ | $2.784 \pm 0.293$ |
| TopoLoss | $0.7621 \pm 0.0036$ | $0.9024 \pm 0.0113$ | $1.083 \pm 0.006$ | $1.076 \pm 0.265$ |
| DMT | $0.7733 \pm 0.0039$ | $0.9077 \pm 0.0021$ | $0.876 \pm 0.038$ | $0.873 \pm 0.402$ |
| Dropout UNet | $0.7410 \pm 0.0019$ | $0.8331 \pm 0.0152$ | $2.013 \pm 0.072$ | $3.121 \pm 0.334$ |
| Prob.-UNet | $0.7429 \pm 0.0020$ | $0.8401 \pm 0.1881$ | $1.873 \pm 0.081$ | $3.080 \pm 0.206$ |
| **Ours** | $\mathbf{0.7814 \pm 0.0026}$ | $\mathbf{0.9109 \pm 0.0019}$ | $\mathbf{0.804 \pm 0.047}$ | $\mathbf{0.767 \pm 0.098}$ |

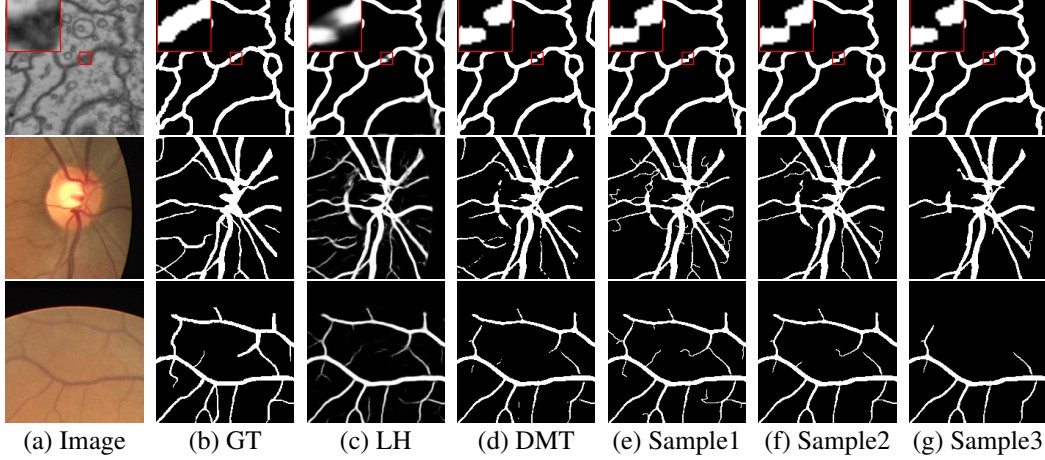

| (a) Image | (b) GT | (c) LH | (d) DMT | (e) Sample1 | (f) Sample2 | (g) Sample3 |

Figure 6: Qualitative results of our method compared to DMT-loss (Hu et al., 2021). From left to right: **(a)** image, **(b)** ground truth, **(c)** continuous likelihood map and **(d)** thresholded binary mask for DMT (Hu et al., 2021), and **(e-g)** three sampled segmentation maps generated by our method.

**Ablation study of loss weights.** We observe that the performances of our method are quite robust to the loss weights $\alpha$ and $\beta$. As the learned distribution over the persistence threshold might affect the final performances, we conduct an ablation study in terms of the weight of KL divergence loss ($\beta$) on DRIVE dataset. The results are reported in Fig. 7. When $\beta = 10$, the model achieves slightly better performance in terms of VOI ($0.804 \pm 0.047$, the smaller the better) than other choices. Note that, for all the experiments, we set $\alpha = 1$.

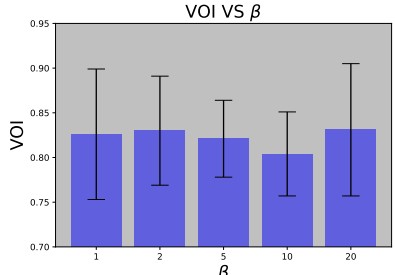

Figure 7: Ablation study for $\beta$.

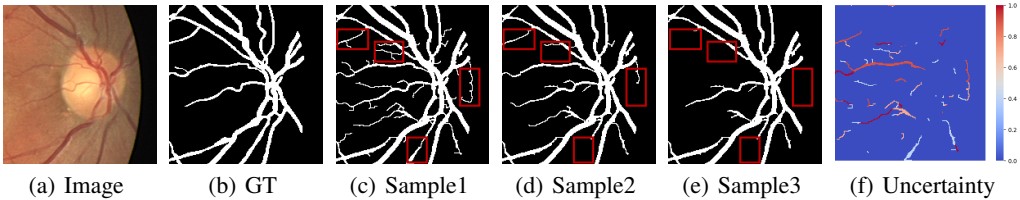

|  (a) Image | (b) GT | (c) Sample1 | (d) Sample2 | (e) Sample3 | (f) Uncertainty |

Figure 8: An illustration of structure-level uncertainty.

**Illustration of the structure-level uncertainty.** We also explore the structure-level uncertainty with the proposed method here. We show three sampled masks (Fig. 8(c-e)) in the inference stage for a given image (Fig. 8(a)), and the structure-level uncertainty map (Fig. 8(f)). Note that in practice, for simplification, different from Sec. 3.2, we empirically generate uncertainty map by taking variance across all the samples (the number is 10 in our case). Different from pixel-wise uncertainty, each small branch has the same uncertainty value with our method. By looking at the original image, we find that the uncertainties are usually caused by the weak signals of the original image. The weak signals of the original image make the model difficult to predict these locations correctly and confidently, especially in structure wise. Actually this also makes sense in real cases. Different from natural images, even experts can not always reach a consensus for biomedical image annotation (Armato III et al., 2011; Clark et al., 2013). More details of structure-level uncertainty are included in Appendix A.1.

**The advantage of the joint training and optimization.** Another straightforward alternative of the proposed approach is to use the discrete Morse theory to postprocess the continuous likelihood map obtained from the standard segmentation networks. More discussions are provided in Appendix A.9.

## 4.2 SEMI-AUTOMATIC EFFICIENT ANNOTATION/PROOFREADING WITH USER INTERACTION

Proofreading is a struggling while essential step for the annotation of images with fine-scale structures. We develop a semi-automatic interactive annotation/proofreading pipeline based on the proposed method. As the proposed method is able to generate structural segmentations (Fig. 8(c)-(e)), the annotators can efficiently proofread one image with rich structures by removing the unnecessary/redrawing the missing structures with the help of structure-level uncertainty map (Fig. 8(f)). The whole inference and structure-aware interactive image annotation pipeline is illustrated in Fig. 5.

We conduct empirical experiments to demonstrate the efficiency of proofreading by using the proposed method. We randomly select a few samples from ISBI13 dataset and simulate the user interaction process. For both the proposed and baseline methods, we get started from the final segmentations and correct *one misclassified branch* each time. For deterministic method (DMT), the user draws one false-negative or erases one false-positive branch for each click. For pixel-wise probabilistic method (Prob.-UNet), the user does the same while taking the uncertainty map as guidance. For the proposed method, the user checks each branch based on the descending order of structure-level uncertainty. VOI is used to evaluate the performances.

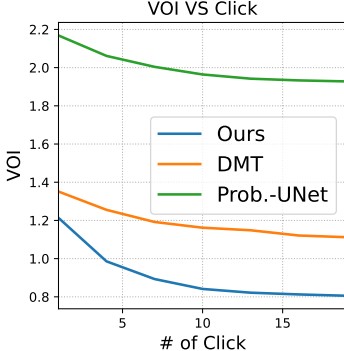

Figure 9: Interaction simulation.

Fig. 9 shows the comparative results of semi-automatic proofreading with user interactions. By always checking branches with highest uncertainty, the proposed method clearly achieves better results and improves the results much faster than baseline methods. Our developed pipeline achieves higher efficiency because of following two perspectives: 1) the generated structural segmentations are essentially topology/structure-preserving; 2) the proposed method provides the structure-level uncertainty map to guide the human proofreading.

## 5 CONCLUSION

Instead of making pixel-wise predictions, we propose to learn structural representation with a probabilistic model to obtain structural segmentations. Specifically, we construct the structure space by leveraging classical discrete Morse theory. We then build a probabilistic model to learn a distribution over structures. In the inference stage, we are able to generate a set of structural segmentations and explore the structure-level uncertainty, which is beneficial for interactive proofreading. Extensive experiments have been conducted to demonstrate the efficacy of the proposed method.

**Reproducibility Statement:** We provide the necessary experimental details in Sec. 4. More specifically, the details of the data are provided in Appendix A.6. The details of baseline methods are described in Appendix A.8. Appendix A.7 contains the evaluations metrics used in this paper. The used computation resources are specified in Appendix A.13.

**Acknowledgement:** The authors thank anonymous reviewers for their constructive feedback. This material is based upon work supported by the Defense Advanced Research Projects Agency (DARPA) under Agreement No. HR0011-22-9-0077. Also, this research of Xiaoling Hu and Chao Chen was partially supported by NSF CCF-2144901. The research of Dimitris Samaras was partially supported by NSF IIS-2212046, IIS-2123920.

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

## A    APPENDIX

Appendix A.1 illustrates structure-level uncertainty compared with traditional pixel-wise uncertainty.

Appendix A.2 shows more qualitative results.

Appendix A.3 provides the details of Morse cancellation.

Appendix A.4 illustrates the details of persistent-homology filtered topology watershed algorithm.

Appendix A.5 describes the reparameterization technique.

Appendix A.6 provides the details of the datasets used in this paper.

Appendix A.7 illustrates the details of the metrics used in this paper.

Appendix A.8 describes the details of the baselines used in this paper.

Appendix A.9 discusses the advantage of joint training and optimization.

Appendix A.10 provides a few instance priors.

Appendix A.11 shows the comparison against noisy likelihood map.

Appendix A.12 illustrates the uncertainty estimation.

Appendix A.13 provides the computational resources for all the conducted experiments.

### A.1    ILLUSTRATION OF THE STRUCTURE-LEVEL UNCERTAINTY

Fig. 1 shows the comparison of traditional pixel-wise uncertainty and the proposed structure-level uncertainty. Specifically, Fig. 1(c) is a sampled segmentation result by Prob.-UNet (Kohl et al., 2018), and Fig. 1(d) is the pixel uncertainty map from Prob.-UNet (Kohl et al., 2018). Different from traditional pixel-wise uncertainty, our proposed structure-level uncertainty (Fig. 1(f)) can focus on the structures.

We also overlay the structure-level uncertainty (Fig. 1(f)) on the original image (Fig. 1(a)), which is shown in Fig. 10. By comparison with the original image, we can observe that the structure-level uncertainty is mainly caused by the weak signals in the original image.

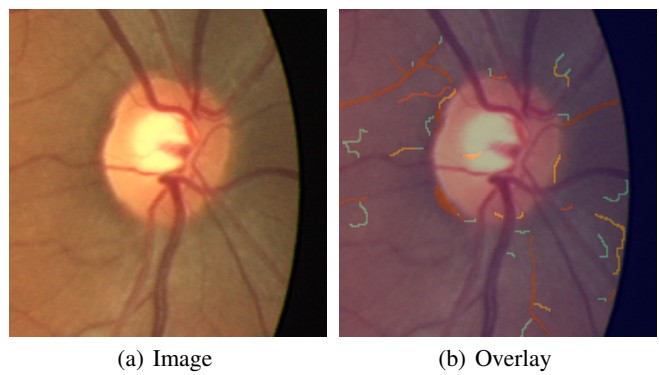

(a) Image                    (b) Overlay

Figure 10: Overlaying the structure-level uncertainty on the original image.

### A.2    QUALITATIVE RESULTS

Fig. 11 shows more qualitative results. From Fig. 11, we can observe that the proposed method can generate both diverse and structure-preserving segmentation maps.

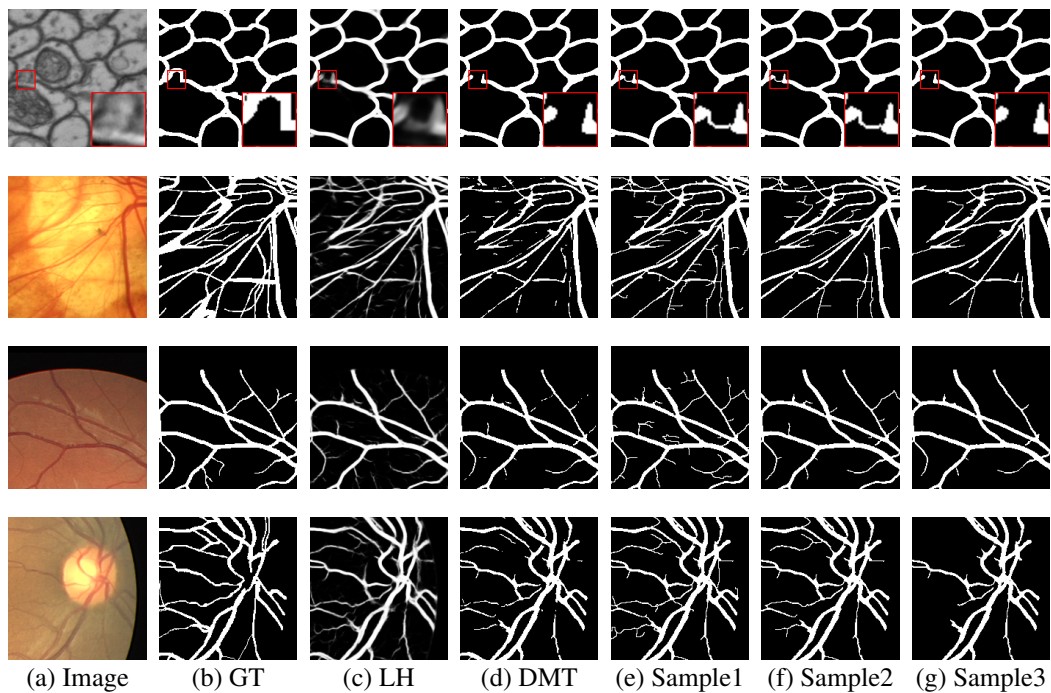

| (a) Image | (b) GT | (c) LH | (d) DMT | (e) Sample1 | (f) Sample2 | (g) Sample3 |

Figure 11: Qualitative results of the proposed method compared to DMT-loss (Hu et al., 2021). From left to right: **(a)** sample image, **(b)** ground truth, **(c)** continuous likelihood map and **(d)** thresholded binary mask for DMT (Hu et al., 2021), and **(e-g)** three sampled segmentation maps generated by our method.

## A.3 ADDITIONAL DETAILS ON THE METHOD

### A.3.1 DISCRETE MORSE THEORY

A 2D image is viewed as a 2-dimensional cubical complex, consisting of 0-, 1-, and 2-dimensional cells corresponding to vertices, edges and squares as its building blocks.

Discrete Morse theory (DMT) (Forman, 1998; 2002) is a combinatorial version of Morse theory for general cell complexes. We will briefly introduce some relevant concepts for the present paper, and describe it in the setting of cubical complexes for images.

*Discrete gradient vector* (also called *a V-pair* for simplicity): Let $K$ be a cubical complex. Given a $p$-cell $\tau$, we denote by $\sigma < \tau$ if $\sigma$ is a $(p-1)$-dimensional face for $\tau$. Discrete gradient vector is a pair $(\tau, \sigma)$ where $\sigma < \tau$.

Given a collection of V-pairs $\mathsf{M}(K)$ over the cubical complex $K$. A sequence of cells $\pi$ : $\tau_0^{p+1}, \sigma_1^p, \tau_1^{p+1}, \sigma_2^p, \cdots, \sigma_k^p, \tau_k^{p+1}, \sigma_{k+1}^p$, where the superscript $p$ in $\alpha^p$ stands for the dimension of cell $\alpha$, form a *V-path* if $(\tau_i, \sigma_i) \in \mathsf{M}(K)$ for for any $i \in [1, k]$ and $\sigma_i < \tau_{i-1}$ for any $i \in [1, k+1]$. A V-path $\pi$ is *acyclic* if $(\tau_0, \sigma_{k+1}) \notin \mathsf{M}(K)$. This collection of V-pairs $\mathsf{M}(K)$ form a *discrete gradient vector field*[2] if (cond-i) each cell in $\mathsf{M}(K)$ can only appear in at most one pair in $\mathsf{M}(K)$; and (cond-ii) all V-paths in $\mathsf{M}(K)$ are acyclic. Given a discrete gradient vector field $\mathsf{M}(K)$, a simplex $\sigma \in K$ is *critical* if it is not in any V-pair in $\mathsf{M}(K)$.

Even though a discrete gradient vector (a V-pair), say $(\tau, \sigma)$ is a combinatorial pair instead of a real vector, it still indicates a "flow" from $\tau$ to its face $\sigma$. A V-path thus corresponds to a flow path (integral line) in the smooth setting. However, to make a collection of V-pairs a valid analog of gradient field, (cond-i) says that at each simplex there should only be one "flow" direction; while

---

[2]We will not introduce the concept of discrete Morse function, as the discrete gradient vector field is sufficient to define all relevation notations.

(cond-ii) is necessarily as flow lines traced by gradient can only go down in function values and thus never come back (thus acyclic).

A critical simplex has "vanishing gradient" as it is not involved in any V-pair in $\mathsf{M}(K)$ (i.e, there is no flow at this simplex). Given a 2D cubical complex $K$ a a discrete gradient vector field $\mathsf{M}(K)$, we can view critical 0-, 1- and 2-cells as minima, saddle points, and maxima, respectively.

Hence, a 1-stable manifold in 2D will correspond to a V-path connecting a critical square (a maximum) and a critical edge (a saddle).

**Morse cancellation.** A given discrete gradient field $\mathsf{M}(K)$ could be noisy, e.g, there are shallow valleys where the mountain ridge around it should be ignored. Fortunately, the discrete Morse theory provides an elegant and purely combinatorial way to cancel pairs of critical simplices (and thus reduce their stable manifolds). In particular, given $\mathsf{M}(K)$, a pair of critical simplices $\langle \delta^{(p+1)}, \gamma^p \rangle$ is *cancellable* if there is a unique V-path $\pi = \delta = \delta_0, \gamma_1, \delta_1, \ldots, \delta_s, \gamma_{s+1} = \gamma$ from $\delta$ to $\gamma$. The *Morse cancellation operation* simple reverse all V-pairs along this path by removing all V-pairs along these path, and adding $(\delta_{i-1}, \gamma_i)$ to $\mathsf{M}(K)$ for any $i \in [1, s+1]$. It is easy to check that after the cancellation neither $\delta$ nor $\gamma$ is critical.

### A.3.2 PERSISTENCE PRUNING

By setting $\rho(\sigma)$ for each cell to be the maximum $\rho$-value of each vertex in $\sigma$, we are able to extend this vertex-valued function $\rho$ to a function $\rho : K \to \mathbb{R}$.

Then how to obtain a discrete gradient vector field from such function $\rho : K \to \mathbb{R}$? Following the approach developed in (Wang et al., 2015; Dey et al., 2018), we initialize a trivial discrete gradient vector field where all cells are initially critical. Let $\epsilon > 0$ be a threshold for simplification. We then perform persistence algorithm (Edelsbrunner et al., 2000) induced by the super-level set filtration of $\rho$ and pair up all cells in $K$, denoted by $\mathcal{P}_\rho(K)$.

Persistent homology is one of the most important development in the field of topological data analysis in the past two decades (Edelsbrunner & Harer, 2022; Edelsbrunner et al., 2000; Zomorodian & Carlsson, 2005). Imagine we grow the complex $K$ by starting from the empty set and gradually include more and more cells in decreasing $\rho$ values, which is the so-called super-level set filtration of $K$ induced by $\rho$. Through this course, new topological feature can be created upon adding a simplex $\sigma$, and sometiems a feature can be destroyed upon adding a simplex $\tau$. Persistence algorithm (Edelsbrunner et al., 2000) will pair up simplices; that is, its output is a set of pairs of simplices $\mathcal{P}_\rho(K) = \{(\sigma, \tau)\}$, where each pair captures the birth and death of topological features during this evolution. The persistence of a pair, say $\mathsf{p} = (\sigma, \tau)$, is defined as $\mathrm{pers}(\mathsf{p}) = \rho(\sigma) - \rho(\tau)$, measuring how long the topological feature captured by $\mathsf{p}$ lives in term of $\rho$. In this case, we also write $\mathrm{pers}(\sigma) = \mathrm{pers}(\tau) = \mathrm{pers}(\mathsf{p})$ – the persistence of a simplex (say $\sigma$ or $\tau$) can be viewed as the importance of this simplex.

With this intuition of the persistence pairings, we next perform Morse-cancellation operation to all pairs of these cells $(\sigma, \tau) \in \mathcal{P}_\rho(K)$ in increasing order their persistence if (i) its persistence $\mathrm{pers}(\delta, \gamma) < \epsilon$ (i.e, this pair has low persistence and thus not important); and (ii) this pair $(\delta, \gamma)$ is cancellable.

Let $\mathsf{M}_\epsilon(K)$ be the resulting discrete gradient field after simplifying all low-persistence critical simplices. We then construct the 1-stable manifolds for the remaining (high persistence, and thus important) saddles (critical 1-cells) from $\mathsf{M}_\epsilon(K)$. Let $\mathcal{S}_1(\epsilon)$ be the resulting collection of 1-stable manifolds. In particular, see an illustration of a V-path (highlighted in black) corresponding to a 1-stable manifold of the green saddle in Fig. 3(b).

### A.4 APPROXIMATION FOR VOLUME DATA

As illustrated in the main text, we propose a persistent-homology filtered topology watershed algorithm to obtain the approximation of Morse structures for volume data. The details are illustrated in Alg. 1.

---

**Algorithm 1:** Persistent-Homology filtered Topology Watershed Algorithm

---

**Input:** a grid 2D image, and a threshold $\theta$
**Output:** Morse structures for volume data
**Definition**: $G = (V, E)$ denote a graph; $f(v)$ is the intensity value of node $v$; lower_star$(v)$ = $\{(u, v) \in E | f(u) < f(v)\}$; $cc(v)$ is the connected component id of node $v$.

1:   PD =$\emptyset$; Build the proximity graph (4-connectivity) for 2D grid image;
2:   $U = V$ sorted according $f(v)$; $T$ a sub-graph, which includes all the nodes and edges whose
     value $< t$.
3:   **for** $v$ in $U$ **do**
4:     $t = f(v), T = T + \{v\}$
5:     **for** $(u, v)$ in lower_star$(v)$ **do**
5:       Assert $u \in T$
6:      **if** $cc(u) = cc(v)$ **then**
6:        Edge_tag$(u, v)$ = loop edge
6:        Continue
7:      **else**
7:        Edge_tag$(u, v)$ = tree edge
7:        younger_cc = $\arg\max_{w=cc(u),cc(v)} f(w)$
7:        older_cc = $\arg\min_{w=cc(u),cc(v)} f(w)$
7:        *pers = t-f(younger_cc)*
8:        **if** pers $>= \theta$ **then**
8:          *Edge_tag$(u, v)$ = watershed edge*
8:          Continue
9:        **end if**
10:       **for** $w$ in younger_cc **do**
10:         $cc(w)$ = older_cc
11:       **end for**
11:       PD = PD + $(f(\text{younger\_cc}), t)$
12:      **end if**
13:     **end for**
14:   **end for**
15:   **return** Membrane_vertex_set = $\cup$ vertices of watershed_edge_set

---

### A.5   REPARAMETERIZATION TECHNIQUE

We adopt the reparameterization technique of VAE to make the network differentiable and be able to backpropagate.

The posterior net randomly draw samples from posterior distribution $\epsilon \sim N(\mu_{post}, \sigma_{post})$. To implement the posterior net as a neural network, we will need to backpropagate through random sampling. The issue is that backpropagation cannot flow through random node; to overcome this obstacle, we adopt the reparameterization technique proposed in (Kingma & Welling, 2013).

Assuming the posterior is normally distributed, we can approximate it with another normal distribution. We approximate $\epsilon$ with normally distribution $Z$ ($Z \sim N(0, \mathbf{I})$).

$$\epsilon \sim N(\mu, \sigma), \qquad \epsilon = \mu + \sigma Z. \tag{3}$$

Now instead of saying that $\epsilon$ is sampled from $Q(X, Y; \omega_{post})$, we can say $\epsilon$ is a function that takes parameter $(Z, (\mu, \sigma))$ and these $\mu$, $\sigma$ come from deep neural network. Therefore all we need is partial derivatives w.r.t. $\mu$, $\sigma$ and $Z$ is irrelevant for taking derivatives for backpropagation.

### A.6   DATASETS

Both volume and vessel datasets are used to validate the efficacy of the proposed method, and the details of the datasets are as follows:

1. *ISBI13* (volume): ISBI13 (Arganda-Carreras et al., 2013) is a EM dataset, containing 100 images with resolution of 1024x1024.

2. *CREMI* (volume): CREMI is another EM dataset, containing 125 images, and each of them has a resolution of 1250x1250.

3. *DRIVE* (vessel): DRIVE (Staal et al., 2004) is a retinal vessel dataset with 40 images. The resolution for each image is 584x565.

We use a 3-fold cross-validation for all the methods to report the numbers over the validation set.

### A.7 EVALUATION METRICS

The details of the metrics used in this paper are listed as follows:

1. *DICE*: DICE score is usually to measure the volumetric overlap between the predicted and ground truth masks.

2. *Adapted Rand Index (ARI)*: ARI is the maximal F-score of the foreground-restricted Rand index (Rand, 1971), a measure of similarity between two clusters.

3. *VOI* (Jain et al., 2010): VOI is a measure of the distance between two clusterings.

4. *Betti Error* (Hu et al., 2019): Betti Error measures the topology difference between the predicted and the ground truth mask. We randomly sample patches over the predicted segmentation and compute the average absolute error between their Betti numbers and the corresponding ground truth patches.

### A.8 BASELINES

We compare the proposed method with two kinds of baselines: 1) Standard segmentation baselines:

1. DIVE (Fakhry et al., 2016) is originally designed for EM data segmentation.

2. UNet (Ronneberger et al., 2015) achieves good performances in different contexts with encoder-decoder scheme and skip connections.

3. UNet-VGG (Mosinska et al., 2018) proposes a topology-aware loss based on the detected linear structures with pretrained filters.

4. TopoLoss (Hu et al., 2019) identifies the topological critical points with persistence homology and derives a novel topological loss.

5. DMT (Hu et al., 2021) identifies the whole topological structures and introduces additional penalty on the whole structures instead of isolated critical points.

2) Probabilistic-based segmentation methods:

1. Dropout UNet (Kendall et al., 2015) dropouts the three inner-most encoder and decoder blocks with a probability of 0.5 during both the training and inference.

2. Probabilistic-UNet (Kohl et al., 2018) introduces a probabilistic segmentation method by combining UNet with a VAE.

For all methods, we generate binary segmentations by thresholding predicted likelihood maps at 0.5.

### A.9 THE ADVANTAGE OF THE JOINT TRAINING AND OPTIMIZATION

As mentioned in the main text, another straightforward alternative of the proposed approach is to use the discrete Morse theory to postprocess the continuous likelihood map obtained from the standard segmentation networks.

In this way, we can still obtain structure-preserving segmentation maps, but there are two main issues: 1) if the segmentation network itself is structure-agnostic, we will not be able to generate satisfactory results even with the postprocessing, and 2) we have to manually choose the persistence threshold to prune the unnecessary branches for each image, which is cumbersome and unrealistic in practice. The proposed joint training strategy overcomes both these issues. First, during the training, we incorporate the structure-aware loss ($L_{skeleton}$). Consequently, the trained segmentation branch itself

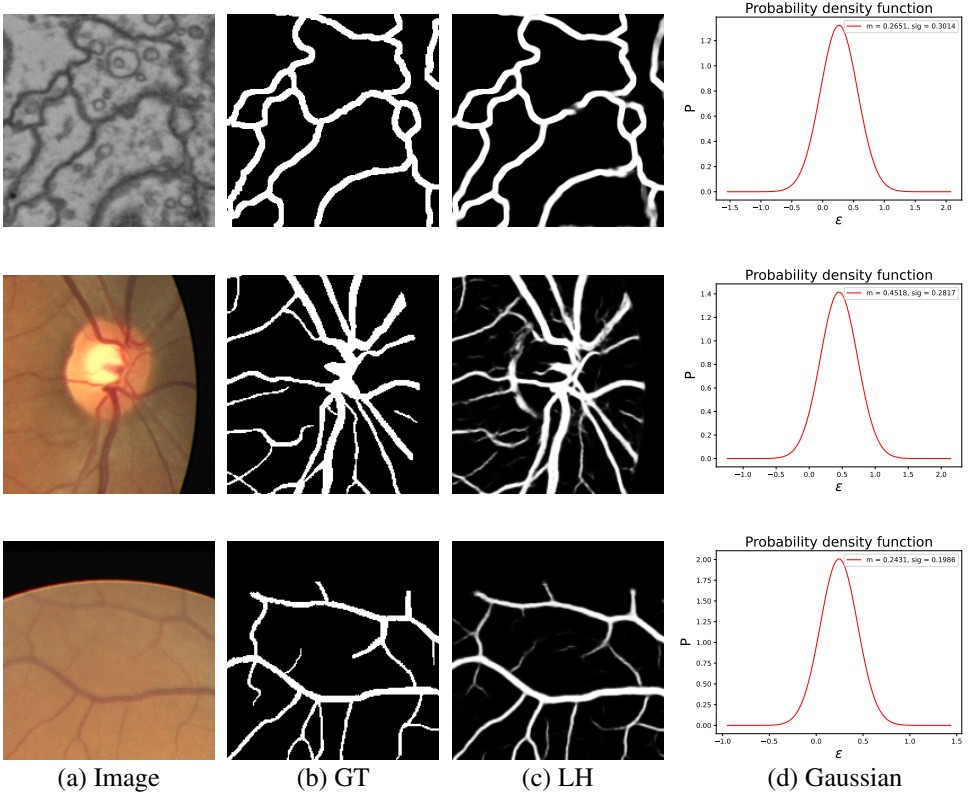

(a) Image  (b) GT  (c) LH  (d) Gaussian

Figure 12: A few instance priors for specific images: **(a)** image, **(b)** ground truth, **(c)** continuous likelihood map and **(d)** Gaussian priors for specific images.

is structure-aware essentially. On the other hand, with the prior and posterior nets, we are able to learn a reliable distribution of the persistence threshold ($\epsilon$) given an image in the inference stage. Sampling over the distribution makes it possible to generate satisfactory structure-preserving segmentation maps within a few trials (the inference will not take long), which is more much efficient.

We conduct an empirical experiment to demonstrate the advantage of the joint training and optimization. For the postprocessing, given the predicted likelihood maps from the standard segmentation networks, we randomly choose five persistence thresholds and generate the segmentation masks separately, and select the most reasonable one as the final segmentation mask to report the performances. The results in Tab. 2 demonstrate the advantage of our joint training and optimization strategy.

Table 2: Quantitative results for comparison of postprocessing on DRIVE dataset.

| Method | Dice ↑ | ARI ↑ | VOI ↓ | Betti Error ↓ |
|---|---|---|---|---|
| Postprocessing | $0.7653 \pm 0.0052$ | $0.8841 \pm 0.0046$ | $1.165 \pm 0.086$ | $1.249 \pm 0.388$ |
| **Ours** | $\mathbf{0.7814 \pm 0.0026}$ | $\mathbf{0.9109 \pm 0.0019}$ | $\mathbf{0.804 \pm 0.047}$ | $\mathbf{0.767 \pm 0.098}$ |

A.10   INSTANCE PRIORS FOR SPECIFIC IMAGES

Here we provide a few instance priors for specific images (Fig. 12).

A.11   ILLUSTRATION OF NOISY LIKELIHOOD MAP

As we know, for image segmentation tasks, with the power of deep neural networks, we are able to achieve quite high pixel-wise segmentation accuracy. In other words, the predicted likelihood maps

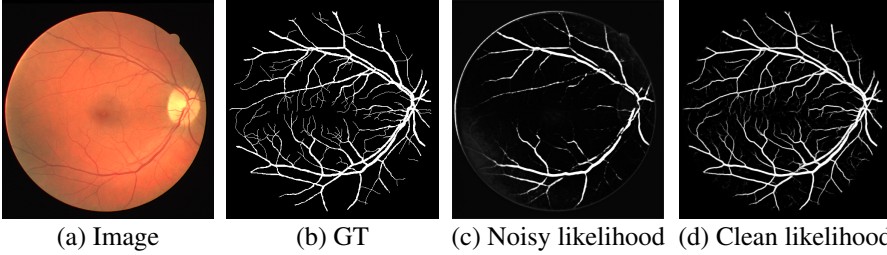

| (a) Image | (b) GT | (c) Noisy likelihood | (d) Clean likelihood |

Figure 13: A comparison of noisy likelihood map: **(a)** image, **(b)** ground truth, **(c)** noisy likelihood map and **(d)** clean likelihood map.

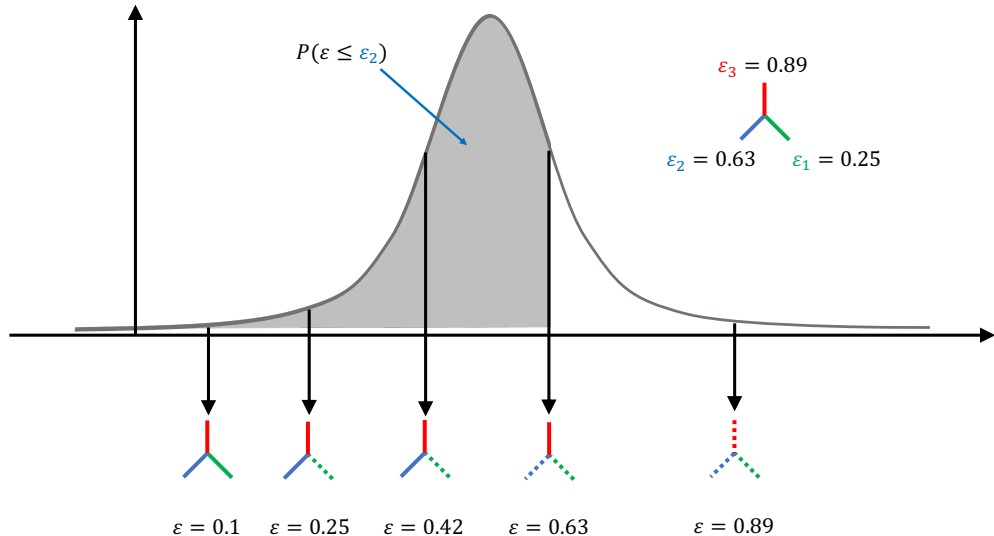

Figure 14: The learned Gaussian distribution of $\epsilon$. We use $\epsilon_1$, $\epsilon_2$ and $\epsilon_3$ to denote the persistence of three different branches shown in green, blue and red, respectively. The shaded regions denotes the probability of branch2 belongs to the final segmentation map $M$.

from the segmentation network are good enough (see Fig. 6(c) and Fig. 11(c)) to distinguish the skeletons/structures of the original images, with issues only happening at locations with weak signals.

Here we provide a comparison of noisy and clean likelihood maps (Fig. 13). If the model is trained to be converged, usually we are able to obtain the clean likelihood map (Fig. 13(d)), and then we could apply discrete Morse theory on it to construct the structural space. On the other hand, if the model is not converged, the obtained likelihood map will be noisy (Fig. 13(c)), and we will not be able to recover the true signals of the original image.

## A.12 Illustration of uncertainty estimation (CDF)

As we mentioned in Sec. 3.2, the probability of a branch $b$ being in a segmentation map $M$ such as $\epsilon_M \sim P$ follows a Bernoulli distribution with the probability $Pr(b)$ being the cumulative distribution function (CDF) of $P$, $CDF_P(\epsilon_b) = P(\epsilon \leq \epsilon_b)$. In Fig. 14, we use dotted lines to denote the filtered out branches and the shaded regions denotes the probability of branch2 belongs to the final segmentation map $M$.

## A.13 Computational resources

All the experiments are performed on a RTX A5000 GPU (24G Memory), and AMD EPYC 7542 32-Core Processor.

