# OpenReview forum: "Learning Probabilistic Topological Representations Using Discrete Morse Theory"
_ICLR.cc/2023/Conference — ICLR 2023 notable top 25%_

### Official Review · Reviewer_PPHt · 2022-10-20

**Confidence:** 5
**Correctness:** 4
**Technical Novelty And Significance:** 3
**Empirical Novelty And Significance:** 3
**Recommendation:** 8

**Clarity, Quality, Novelty And Reproducibility:**

## Clarity

- I stumbled over some of the claims in the abstract: specifically, since the paper already cites several related methods that employ topological ideas (Hu et al., 2019, for instance), the sentence "first deep learning base method [...]" in the abstract should be rephrased. Similar constructions happen at other places. I understand the delineation to previous work, and I definitely see the merits of this paper; hence, in my opinion, such statements are not necessary (and in fact, the related work section provides a more objective delineation that is sufficient for me).

- If "structural" is to be used as a synonym for "topological" throughout the paper, I would already remark on this in the abstract. As it stands right now, I was a little bit confused about the meaning of this.

- Would it be possible to highlight the improvements in Figure 1 better? I liked the idea of combining the segmentation map with the original data, as showcased in the appendix. I think Figure 1 would benefit from a similar treatment.

- Figure 2 (and its explanation) might be used as segue into explaining the impact of topological defects.

- When mentioning the quality of the method at the beginning of Section 3, please extend and formalise the aspect on being "close enough to the true structures".

- Please elaborate on the meaning of a "reasonably clean input". This reads vague at the first time and requires careful reading to fully understand.

- The point about extending this from 2D to 3D is repeated multiple times. I think it would be sufficient to discuss this in *one* place. Personally, I am also fine with the paper focusing only on the 2D case for now; that case is already sufficiently complex and interesting on its own!

- When explaining some of the "moving parts" of the method in Section 3.1, I would appreciate a brief discussion of how the "filling" of *partially observed* structures works in practice. The example in Figure 3 is really captivating, and it would present a great opportunity to better comment on this aspect.

- Consider adding some explanations about cubical complexes or citing related work in this area; it might not be familiar to all readers (in general, I would say that the paper already does an excellent job at presenting the necessary background information).

- The section on uncertainty quantification is missing a justification as to where the Bernoulli distribution is arising from. Is this an approximation type argument?

- When showing Figure 8, can some of the differences between images be highlighted better? Why not pick several structures that are differing the most between the samples?

## Quality

While the paper is already of high quality, making a strong, substantial contribution, there are several aspects that could be commented on in addition:

- Having described the method, what are its limitations? Can the method capture all relevant structures? Are there thing it is not particularly adept at (for instance, segmentations whose ground truth has topological defects)? Are there additional theoretical guarantees that could be highlighted?

- Please refer to the pruning procedure on page 5 when discussing Figure 2. Some of the high-level details from Appendix A.3 should also be added to to this section in order to make the main text self-contained.

## Novelty

The method is novel in that it provides a way to traverse the space of potential segmentations, making use of discrete Morse theory and persistent homology to generate appropriate structures. Delineation to existing work is done to a sufficient extent, and it is clear to me that the proposed method constitutes a substantial contribution to the field.

## Reproducibility

Concerning this aspect, I am on the fence: on the one hand, the paper describes the method in sufficient detail; on the other hand, additional code as part of the supplementary materials would have been appreciated. Nevertheless, I think that a skilled graduate student should be able to reproduce the bulk of the method.

## Minor issues

The paper is already written very well, and it was a joy to read and comment on! There are some minor issues with the text that require another pass, including some missing articles. Here are some other minor issues I found:

- The bibliography is suffering from some inconsistencies; please check capitalisation and author names again. For instance, it should be "Morse theory", not "morse theory".

- "existing [...] map is" --> "existing maps [...] are"

- "When it comes down to" --> "When it comes to"

- "an 1D" --> "a 1D"

- "an 1-stable" --> "a 1-stable"

- "all possible combination" --> "all possible combinations"

- "can't" --> "cannot" (admittedly, a personal preference)

- "different $\epsilon$'s" --> "different $\epsilon$ values"

- "changing its topology" --> "changing their topology" (I think the reference here is to *skeletons*)

- "especially in structure wise": I have trouble parsing this sentence.

- "struggling": maybe "strenuous" would be more appropriate?



**Strength And Weaknesses:**

This paper presents a strong high-quality contribution for obtaining topological representations of 2D and 3D images, which in turn can be used to perform downstream tasks, such as segmentation. The main strengths of the paper lie in its **firm theoretical foundation** and its **thorough experimental setup**. There are only minor issues to be rectified in the present write-up, mostly concerning the clarity of the approach: to make the paper fully accessible to an audience that may not be entirely familiar with all the concepts, some additional modifications of the text (see below) are required. Moreover, certain claims in the paper can be slightly rephrased. That being said, the paper at hand has no substantial weaknesses that would prevent publication. On aspect that slightly marred the review for me was that no code is being provided; this does not detract from reproducibility, though.


**Summary Of The Paper:**

This paper presents a new method for learning topological representations of 2D and 3D images. In contrast to existing work, the paper proposes a method that leverages the intrinsic structure of the "representation space", making it possible to derive (a) topological representations in a probabilistic fashion, while (b) also enabling and facilitating the analysis of structural uncertainty. The new method is then used to to primarily address segmentation task, exhibiting high-quality performance in different data sets.


**Summary Of The Review:**

This paper was a joy to read; it provides a substantial contribution and, for the most part, presents all results and concepts in a very clear fashion. The clarity issues that I listed can be easily addressed by some restructuring and rewriting within the revision cycle. I am thus happy to provisionally endorse this paper for publication.

---

> ### Author Response · Authors · 2022-11-19
> **Response to the comments of Reviewer PPHt (1/2)**
>
> We sincerely thank you very much for your detailed comments. We really appreciate your efforts to help improve the quality of this manuscript. We have fixed all the language issues according to your comments. Below we address other specific concerns one-by-one.
>
> **Q1**: I stumbled over some of the claims in the abstract.
>
> **A1**: Thanks very much for your suggestion. We have revised our reference to previous work per your suggestions.
>
> **Q2**: If "structural" is to be used as a synonym for "topological" throughout the paper, I would already remark on this in the abstract.
>
> **A2**: We have fixed this in the revised version.
>
> **Q3**: Would it be possible to highlight the improvements in Figure 1 better?
>
> **A3**: We have updated Fig. 1 as you suggested, particularly  have overlayed the structure-wise uncertainty map on the original image (Fig. 1(f)). We will appreciate any additional presentation suggestions you might have.
>
> **Q4**: When mentioning the quality of the method at the beginning of Section 3, please extend and formalise the aspect on being "close enough to the true structures". Please elaborate on the meaning of a "reasonably clean input".
>
> **A4**: As we know, for image segmentation tasks, with the power of deep neural networks, we are able to achieve quite high pixel-wise segmentation accuracy. In other words, the predicted likelihood maps from the segmentation network are good enough (see Fig. 6(c) and Fig. 11(c)) to distinguish the skeletons/structures of the original images, with issues only happening at locations with weak signals. We have added this explanation to the Appendix Section A.11.
>
> We agree that ‘’reasonable cleanliness’’ is in the eye of the beholder. Instead of a formal definition, we have provided examples of what we consider noisy and clean likelihood maps in the Appendix Section A.11 (Fig. 13).
>
> **Q5**: The point about extending this from 2D to 3D is repeated multiple times.
>
> **A5**: We have fixed this as you suggested.
>
> **Q6**: When explaining some of the "moving parts" of the method in Section 3.1, I would appreciate a brief discussion of how the "filling" of partially observed structures works in practice. The example in Figure 3 is really captivating, and it would present a great opportunity to better comment on this aspect.
>
> **A6**: We focused on the 1-stable manifolds of 2D images in this paper. As illustrated in Fig. 3, if we view the likelihood map as a terrain function, the Morse structure captures the mountain ridge, which will recover the complete branch even through the low-signal region. More technical details could be found at Appendix Section A.3 of the updated version.
>
> **Q7**: Consider adding some explanations about cubical complexes or citing related work in this area; it might not be familiar to all readers.
>
> **A7**: Thanks for the nice suggestion! We have added a few references when first introducing the cubical complexes in the ‘Discrete Morse theory’ section (Section 3.1).
>
> **Q8**: The section on uncertainty quantification is missing a justification as to where the Bernoulli distribution is arising from. Is this an approximation type argument?
>
> **A8**: This comes from the structural space construction and it is exact. Recall we have a one-parameter family of sub-complexes parameterized by persistence. A branch appears in the sub-complex if and only if its associated persistence is higher than the sampled threshold. We learn a prior distribution $P$ over the persistence threshold $\epsilon$. So the chance of a particular branch appearing in the sub-complex is the chance of the sampled persistence threshold is below its associated persistence, which is the CDF of $P$.
>
> We have added an illustration of uncertainty estimation for branches in Fig. 14 (Section A.12 of Appendix) in the revised version.
>
> **Q9**: When showing Figure 8, can some of the differences between images be highlighted better? Why not pick several structures that are differing the most between the samples?
>
> **A9**: We have updated Fig. 8 to highlight the differences between images as you suggested, by picking several structures.
>
> **Q10**: Having described the method, what are its limitations?
>
> **A10**: Our work directly learns topological representations from images. Despite the novel approach, we do feel pruning the structures with a global persistence threshold is somewhat unsatisfying. Ideally, we should be able to adaptively prune the unlikely branches locally. Another issue is the likelihood map and its induced structures may not be perfect. This could be potentially improved in a smarter way. These are left for further work.

---

> > ### Author Response · Authors · 2022-11-19
> > **Response to the comments of Reviewer PPHt (2/2)**
> >
> > **Q11**: Can the method capture all relevant structures? Are there things it is not particularly adept at (for instance, segmentations whose ground truth has topological defects)? Are there additional theoretical guarantees that could be highlighted?
> >
> > **A11**: These are all excellent questions. As discussed above, the ability of our method is constrained by the underlying likelihood function. If the likelihood function is poor and misses some structures, it will be missed from the structural space, and will not be captured by the model. As to topological defects in the ground truth, it will be worth evaluating. But we hypothesize that our probabilistic model will be more robust to such defects compared to prior topology-aware loss (e.g., Hu et al. 2019). The reason is even if a branch is mistakenly broken in the ground truth, our model will reason about the whole branch and tends to give it a high probability. Finally, theoretical guarantee is possible. It will require assumptions about the underlying noise model (as discussed in Q3 of Reviewer Jetp), about the signal noise ratio, and about the geometry of the branches (thickness, intensity, etc).
> >
> > **Q12**: Please refer to the pruning procedure on page 5 when discussing Figure 2.
> >
> > **A12**: We have fixed this as you suggested.
> >
> > **Q13**: Some of the high-level details from Appendix A.3 should also be added to this section in order to make the main text self-contained.
> >
> > **A13**: Thanks for your suggestion. Due to space limitations, we might be missing details that reduce readability of this section. Depending on additional corrections we need to make in the final version, we will try to add more details if space permits.
> >
> > **Q14**: Additional code as part of the supplementary materials would have been appreciated.
> >
> > **A14**: We will release the code upon acceptance.
> >
> > **Q15**: There are some minor issues with the text that require another pass, including some missing articles.
> >
> > **A15**: We have incorporated your suggestions in the revised version. Thank you very much for pointing them out.
> >
> > Thank you again for all your suggestions to improve our manuscript. If you have further suggestions/thoughts, please kindly let us know. We will be happy to improve the manuscript accordingly.

---

> > > ### Comment · Reviewer_PPHt · 2022-11-21
> > > **Thanks for a comprehensive answer**
> > >
> > > I thank the authors for their comprehensive answers to my queries—I will use this in my deliberations with my reviewer colleagues now.

---

> > > > ### Author Response · Authors · 2022-11-21
> > > > **Thanks for your response**
> > > >
> > > > Dear Reviewer PPHt,
> > > >
> > > > Thanks very much for your response! Again, thanks for your valuable comments to improve our manuscript.
> > > >
> > > > Best,
> > > >
> > > > Authors of paper #4288

---

### Official Review · Reviewer_Jetp · 2022-10-29

**Confidence:** 4
**Correctness:** 4
**Technical Novelty And Significance:** 3
**Empirical Novelty And Significance:** 2
**Recommendation:** 6

**Clarity, Quality, Novelty And Reproducibility:**

The idea is novel.
Paper is well written and clearly contrasts the strengths of the work against other baselines used.
The description in the paper seems adequate to reproduce the experiments.  There is no statement about that the code will be made available for ease of reproducibility.

**Strength And Weaknesses:**

Key strengths:
The deep architectural formulation and integration of ideas from morse theory and persistence homology is interesting.

Weaknesses:
The topic of vessel segmentation or topological structure estimation from 2D/3D data is something that has been studied in the literature for several decades.  While the paper claims that the new method outperforms other baselines, it appears that the new technique's performs is very close in performance to that of DMT method.  The impact of the connectivity improvements on end application goals must be described in order to really gauge in what concrete scenarios the DMT method was outperformed.    I feel that explicit knowledge of the noise characteristics of the sensing modality are critical for adaptively rank ordering possible branch hypotheses (for a given input image instance) in low signal to noise conditions where the improvements are being sought.  While I understand the rationale for the prior network and the KL divergence loss between the prior and posterior network parameter estimates,  it will be helpful to see in the appendix details of how well the instance specific priors look for specific images.




**Summary Of The Paper:**

The focus of the paper is on extraction of structures like vessels in 2D, 3D image datasets.  The central idea in the paper is a deep learning algorithm that can deal with topological representations.  Topological structure variability is modeled by a one-parameter family facilitated by discrete Morse theory and persistent homology.  Unlike other schemes that deal with pixelwise probabilities for describing segmentation variability, this paper learns a probabilistic model that performs inference directly in topological (structural) representation space.  Results are shown to illustrate that the new method outperforms other methods that rely on pixelwise probabilities. Moreover, the utility of the uncertainty estimates on the topological structures for ease of editing is illustrated.

**Summary Of The Review:**

Overall I find the paper quite interesting and feel it has value. The main concern I have is on the justification of the empirical claims and in having more details on the inner working of the devised pipeline (please see weakness section).

---

> ### Author Response · Authors · 2022-11-19
> **Response to the comments of Reviewer Jetp**
>
> Thanks for your constructive feedback. Below we address specific concerns one-by-one.
>
> **Q1**: Justification of the empirical claims. It appears that the new technique’s performs are very close in performance to that of DMT method.
>
> **A1**: The improvement of our method is statistically significant. In Table 1, we used a t-test (95% confidence interval) to determine the statistical significance and highlight the significantly better results. Our method performs significantly better than others (including DMT) in terms of topology-aware metrics (ARI, VOI and Betti Error). As we have also pointed out in Q7 for reviewer 4b8b, the quantitative results also show that our proposed method achieves better/comparable DICE numbers (a non topology-aware metric) compared to strong baselines.
>
> More importantly, a major contribution of this paper is that we perform direct inference on structures, instead of pixel-wise predictions. This is in contrast to existing topology-aware methods, including DMT. Our method learns a probability model in the structural space and generates true structures rather than pixel-maps. This will (1) lead to better topological integrity in automatic segmentation tasks (as shown in Table 1); and (2) facilitate semi-automatic interactive annotation/proofreading via the sampling of structures and structure-aware uncertainty (see Section 4.2).
>
> **Q2**:  I feel that explicit knowledge of the noise characteristics of the sensing modality are critical for adaptively rank ordering possible branch hypotheses (for a given input image instance) in low signal to noise conditions where the improvements are being sought.
>
> **A2**: This is a very good question! Yes, you are right! Structural reasoning should use the underlying noise model and noise level. In this paper, we only focused on modeling the uncertainty of branches at the image analysis level. Our uncertainty estimation is based on the expert annotation, which we consider reasonably reliable. It will be very exciting and impactful future work to expand the current probabilistic model to incorporate the noise model of image acquisition and image reconstruction (which is of course modality-dependent).
>
> **Q3**: It will be helpful to see in the appendix details of how well the instance specific priors look for specific images.
>
> **A3**: Good suggestion! We have provided a few samples in the Appendix Section A.10 (Fig. 12).
>
> **Q4**:  There is no statement about that the code will be made available for ease of reproducibility.
>
> **A4**: We will release the code upon acceptance.

---

> > ### Comment · Reviewer_Jetp · 2022-11-21
> > **Thank you for the clarifications**
> >
> > Thank you for the clarifications!  I will weigh in your feedback during the internal discussions with other reviewers.

---

> > > ### Author Response · Authors · 2022-11-21
> > > **Thanks for your response**
> > >
> > > Dear Reviewer Jetp,
> > >
> > > Thanks very much for your response! Please kindly let us know if you have further questions. We will be happy to address your concerns.
> > >
> > > Best,
> > >
> > > Authors of paper #4288

---

### Official Review · Reviewer_4b8b · 2022-10-31

**Confidence:** 2
**Correctness:** 3
**Technical Novelty And Significance:** 3
**Empirical Novelty And Significance:** 2
**Recommendation:** 6

**Clarity, Quality, Novelty And Reproducibility:**

The novelty of the concepts and methods is recognized, but there are doubts about the clarity of the description of the methods. This makes it difficult to fully assess the quality of the methods.

**Strength And Weaknesses:**

Strength

- We raised the issue of conventional segmentation technology and created a method to solve it.
- Experiments show that the proposed method significantly outperforms the conventional method

Weakness

- Before the substantive discussion, one major concern is that this paper is a difficult text to understand in terms of the proposed methodology. One of the reasons for this is the use of uncommon terms and symbols without explanation, although the terminology may be common knowledge among the authors. It is difficult to present all of them, but the following are representative ones.
	- In P6L1, Y remains undefined. A short time later, Y is shown to be ground truth, but it is extremely difficult to read. Also, not a big deal, but $L_{bce}$ is marked as cross-entropy, but I don't know what bce stands for.
	- In equation (1), $Y\circ S$ is written, but I don't know what it means.
	- The terminology regarding TDA, persistent homology, seems to be different from the common one. A persistent diagram in persisitent homology is one that extracts changes in the features of a figure that vary with filtration by a certain parameter. The proposed PD seems to refer to a method to create a figure (a single figure) by connecting point clouds when the parameter is a certain value. The proposed method is a major obstacle to understanding the correction by the simplex construction and its filtration, which seems to play an important role.
- This method seems to clarify the segment boundaries and compare them with the ground truth while adjusting the degree of clarification. I don't understand why ground truth Y and $S_{skelton}$ are combined in (1) with the above understanding, partly because the meaning of $Y\circ S$ is not clear. Also, in fig.1, Y and $S_{skelton}$ still do not seem to be combined, which is inconsistent with equation (1). This part should clarify definitions and implications.
- I assume that by comparing skelton and ground truth with well-defined boundaries, we are trying to make the topological information, i.e., the information of segmented chunks, similar, but why use cross-entropy for this purpose? I think you are looking at pixel-by-pixel information after all. I think there is a way to get more chunky information on a Morse theory basis. A simple idea is the Weierstrass distance of a persistent diagram (in the general sense, not in the sense of this paper) using Morse theory-based sub-level filtration, for example.
- The results shown in the experiment are an evaluation metric for topological information. This is not surprising, since the proposed method takes topology information into account, whereas conventional methods do not. If these indices are more valid than conventional indices for segmentation evaluation, they should be discussed. Alternatively, a comparison should be made with the evaluation indices provided by conventional methods.

**Summary Of The Paper:**

This paper proposes a novel technique for segmentation, especially for data with fine-scale structure. The authors questioned the conventional segmentation technique, which evaluates the segmentation on a pixel-by-pixel basis and overlooks the perspective of topological structure, and proposed a method to solve this problem. The proposed method proposes a Deep Learning structure that introduces a method for handling topological data, and compares its effectiveness with a typical baseline method.

**Summary Of The Review:**

It is interesting to point out the problems with conventional segmentation methods and to propose methods to solve them. On the other hand, it is difficult to judge the value of the proposed method because it is unclear why the issue is important and there are many parts of the text that are unclear, such as the proposed method. At the very least, the sentence structure should be reviewed.

**Conclusion following discussion with the authors**

The problem of sentence structure, which hinders understanding of the proposed method, has been greatly improved. There is still room for improvement of the method, but it does provide some progress on the segmentation problem. I recognize that this is a positive result, and I will raise my score.

---

> ### Author Response · Authors · 2022-11-19
> **Response to the comments of Reviewer 4b8b (1/2)**
>
> Thanks very much for your constructive feedback! We appreciate your questions as they point out potential gaps in our explanations. We have tried to clarify and improve our manuscript accordingly. We will address any remaining clarity issues in the final copy. The changes are highlighted in blue in the revised manuscript.
>
> *(Clarification)*
>
> **Q1**: In P6L1, $Y$ remains undefined. A short time later, $Y$ is shown to be ground truth.
>
> **A1**: As you mentioned, $Y$ is denoted as the ground truth in the whole paper, which is a common notation. In the revised version, we have added the explanation for $Y$ just after the formula.
>
> **Q2**: Also, not a big deal, but $L_{bce}$ is marked as cross-entropy, but I don't know what bce stands for.
>
> **A2**: $bce$ denotes the binary cross-entropy. We have made it more specific in the revised version.
>
> **Q3**: In equation (1),  $Y \circ S$ is written, but I don't know what it means.
>
> **A3**: As we explained in the main text just before Eq. 1, $Y \circ S$ denotes the *skeleton-masked ground truth*. A pixel/voxel will retain its value in the ground truth ($Y$) if it is inside the skeleton mask ($S$). Otherwise its value is zero. Here $\circ$ is the Hadamard product (element-wise product). We have added the explanations in the revised version.
>
> **Q4**: The terminology regarding TDA, persistent homology, seems to be different from the common one.
>
> **A4**: Thank you for pointing out the confusion. The necessary information is in the manuscript, however, due to space constraints, the explanation is terse. The persistent homology terminology we used is standard. Here the persistent homology is based on the classic super-level set filtration of the image using the likelihood function as the filter function [1].
>
> Here we provide additional clarifications. In our paper, we use the discrete Morse theory. There is a close relationship between the discrete Morse complex and persistent homology. The discrete Morse complex contains all critical points which create/destroy persistent homology classes. A Morse structure connects a pair of critical points which create and destroy a persistent homology class (a point in the persistence diagram). Thus we can use the persistence of the corresponding class to weight this Morse structure. If we update the function values along the Morse structure (an operation called Morse cancellation), we can effectively remove the corresponding persistent point in the persistence diagram. More details on persistence-based pruning of Morse structures can be found in [2,3]. We have added more details in the supplementary section Appendix Section A.3.
>
> [1] Wagner, H., Chen, C. and Vuçini, E., Efficient computation of persistent homology for cubical data. In Topological methods in data analysis and visualization (TopoInVis), 2011
>
> [2] S. Wang, Y. Wang, and Y. Li. Efficient map reconstruction and augmentation via topological methods. In Proc. 23rd ACM SIGSPATIAL, pp. 25. ACM, 2015.
>
> [3] Tamal K. Dey, Jiayuan Wang, and Yusu Wang. Graph reconstruction by discrete morse theory. In
> Proc. 34th Intl. Sympos. Computational Geom. (SoCG), pp. 31:1–31:15, 2018
>
> **Q5**: I don't understand why ground truth $Y$ and $S_{skeleton}$ are combined in (1) with the above understanding. Also, in fig.1, $Y$ and $S_{skeleton}$ still do not seem to be combined, which is inconsistent with equation (1).
>
> **A5**: Thank you for pointing this out. We have clarified this in the paper. As explained in Q3, $Y \circ S$ denotes the *skeleton-masked ground truth*. We essentially use the skeleton to mask both the ground truth Y and the likelihood map, and compare them with binary cross-entropy loss. In other words, we are comparing the likelihood map and the ground truth only within the skeleton mask. We have updated Fig. 4 for illustration. Note that this comparison only happens at the training stage and is used to measure the quality of a sampled skeleton. At the inference stage, the model output will simply be a sample skeleton, expanded with segmentation based on the likelihood function (as in Fig. 1(e) and Fig. 2(f)). See the bottom of page 6 for more details ('*Inference stage: generating structure-preserving segmentation maps.*').

---

> > ### Author Response · Authors · 2022-11-19
> > **Response to the comments of Reviewer 4b8b (2/2)**
> >
> > **Q6**: But why use cross-entropy for this purpose? I think you are looking at pixel-by-pixel information after all. I think there is a way to get more chunky information on a Morse theory basis. A simple idea is the Weierstrass distance of a persistent diagram (in the general sense, not in the sense of this paper) using Morse theory-based sub-level filtration.
> >
> > **A6**: This is a good question so let us clarify. We used binary cross-entropy loss in two places, the segmentation loss (Eq. 2) and the skeleton loss (Eq. 1). The segmentation loss is to ensure the likelihood map is reasonably good. Extracting and reasoning about the Morse complex requires a reasonable likelihood map as the filter function to start with.
> >
> > As for the skeleton loss (Eq. 1), doing a pixel-wise comparison instead of structure-level comparison is inevitable. The fundamental issue is that the ground truth is not given at structure-level ($Y$ is a binary map on all pixels). Our skeleton loss has to compare the predicted structures/skeletons with the ground truth at every pixel. We use skeleton-masked binary cross-entropy loss so that we can incorporate the pixel-wise information coming from the neural network; if the neural network is very confident/unconfident at particular pixels within a structure, this could be taken into account. A possible alternative is to decompose the ground truth into structures and compare them with predicted structures piece-wise. We’d be happy to try this route in the future, keeping in mind that such a heuristic decomposition may introduce artifacts.
> >
> > Directly comparing persistence diagrams with Wasserstein distance (we assume this is what you meant) can be useful when we only want to match the prediction and the ground truth in Betti number (as in Hu et al. 2019). The downside is that the matching induced in the Wasserstein distance is purely based on birth/death times. It loses the natural matching between a predicted Morse structure and its counterpart in the ground truth, which is needed for structure-level supervision.
> >
> > **Q7**: If these indices are more valid than conventional indices for segmentation evaluation, they should be discussed. Alternatively, a comparison should be made with the evaluation indices provided by conventional methods.
> >
> > **A7**: We totally agree that evaluating on conventional metrics is important. In our Table 1 column 1, we did report the DICE score. It is one of the most popular pixel-wise evaluation metrics for segmentation. The quantitative results show that our proposed method achieves better/comparable DICE numbers compared to strong baselines. The second and third metrics (ARI and VOI) are topology-aware, but are also classic metrics for edge detection [4].
> >
> > [4] P. Arbelaez, M. Maire, C. Fowlkes and J. Malik., “Contour Detection and Hierarchical Image Segmentation”, IEEE TPAMI, Vol. 33, No. 5, pp. 898-916, May 2011.
> >
> > As a final note, the fourth metric, Betti error, focuses on topological accuracy. Since its introduction in 2019, it has been quickly adopted by the literature as the significance of topology in downstream tasks is evident (Hu et al., 2019, Hu et al., 2021, Shit et al., 2021).
> >
> > Thank you again for your constructive review. We hope we have addressed all your concerns. If you have any further questions, please kindly let us know. We are happy to discuss further.

---

> > > ### Comment · Reviewer_4b8b · 2022-11-19
> > > **Thanks for the clarification**
> > >
> > > Hi, Authors,
> > >
> > > Thanks for the clarification and revision.
> > >
> > > Thank you for the clarification to my concerns and the revision of the manuscript. I have clarified and agreed with you regarding A1-5 and 7.
> > > Regarding A6, I too understand that the simple use of the Weierstrass distance does not work. On that basis, I understood this time that you proposed a method that works as a first step in this case. I look forward to further discussion on this point.
> > >
> > > Basically, I believe I can recommend this paper, but will double check and revise the score.
> > >
> > > Best,

---

> > > > ### Author Response · Authors · 2022-11-19
> > > > **Glad that you agree with our clarifications/explanations**
> > > >
> > > > Dear Reviewer 4b8b,
> > > >
> > > > Thanks very much for your quick response and we are glad that you agree with our clarifications/explanations.
> > > >
> > > > We are looking forward to discussing with you further.
> > > >
> > > > Best,
> > > >
> > > > Authors of paper #4288

---

> > > > > ### Author Response · Authors · 2022-11-24
> > > > > **Follow-up**
> > > > >
> > > > > Dear Reviewer 4b8b,
> > > > >
> > > > > Do you still have any concerns regarding Q6/A6? We will be happy to discuss further and address your concerns if there are any.
> > > > >
> > > > > Best,
> > > > >
> > > > > Authors of paper #4288

---

### Author Response · Authors · 2022-11-19
**General response**

We thank all the reviewers for their valuable feedback! We are encouraged that all reviewers appreciated the novelty of the contribution. We have improved our presentations according to the suggestions. We have updated the revised version to reflect the modifications (highlighted in blue).

Below we address specific concerns one-by-one.

---

### Public Comment · ~Yuri_Smirnov1 · 2023-02-28
**Missing citation**

Thank you for the article. Sadly, again a comment on the missing citation. Since you are using the persistence pairing, here is the reference for the paper where it was first introduced, under the name of "canonical form" : Barannikov, S.(1994) "The Framed Morse Complex and its Invariants", Advances in Soviet Mathematics, 21: 93–115. Also, the algorithm for the computation of the persistence pairing, called "the persistence algorithm" on page 5 in your article, is described in section 2.1 in the loc.cit. paper.

---

### Decision · Program_Chairs · 2023-01-20

**Decision:**

Accept: notable-top-25%

**Justification For Why Not Higher Score:**

The enthusiasm from this work exists but somehow limited. Can be bumped to spotlight.

**Justification For Why Not Lower Score:**

Solid and novel framework for learning the threshold of persistence homology in 2D images. Reviewers unanimous for accepting this work.

**Metareview: Summary, Strengths And Weaknesses:**

There is a consensus among reviewers to accept this work. All reviewers appreciated the incorporation of topological tools into segmentation learning setup. Reviewers commented about some exposition issues that were addressed in the rebuttal in a satisfactory manner. Other concerns raised are extent of improvement over existing vessel/topological segmentation methods,  but that does not seem as a roadblock to acceptance. We encourage the authors to publish code for reproducibility and facilitate future work on this topic.

**Note From Pc:**

if the above contains the word "oral" or "spotlight" please see: "oral" presentation means -> notable-top-5% and "spotlight" means -> notable-top-25%. As stated in our emails, we are disassociating presentation type from AC recommendations